Metagenomic and satellite analyses of red snow in the Russian Arctic

Hisakawa Nao 1
Quistad Steven D. 1
Hester Eric R. 1 2
Martynova Daria 3 4
Maughan Heather 5 heathermaughan@gmail.com
Sala Enric 6
Gavrilo Maria V. 4
Rohwer Forest 1 frohwer@gmail.com
1 Department of Biology, San Diego State University , San Diego, CA , United States
2 Department of Microbiology, Radboud University Nijmegen , Nijmegen , Netherlands
3 White Sea Biological Station, Zoological Institute, Russian Academy of Sciences , St. Petersburg , Russia
4 National Park Russian Arctic , Archangelsk , Russia
5 Ronin Institute , Montclair, NJ , United States
6 National Geographic Society , Washington, DC , United States
Bishop-Lilly Kimberly
Electronic publication date: 2015 Dec 10
Publication date: 2015
Volume: 3
Electronic Location ID: e1491
Received 2015 Sep 3; Accepted 2015 Nov 19
Copyright: © 2015 Hisakawa et al.
Copyright year: 2015
Copyright holder: Hisakawa et al.
License: This is an open access article distributed under the terms of the Creative Commons Attribution License, which permits unrestricted use, distribution, reproduction and adaptation in any medium and for any purpose provided that it is properly attributed. For attribution, the original author(s), title, publication source (PeerJ) and either DOI or URL of the article must be cited.
License URL: https://creativecommons.org/licenses/by/4.0/

Keywords: Red snow, Snow, Arctic, Watermelon snow, Viruses, Franz Josef Land, Phage, Metagenomics

Funding: Russkaya Arktika (Russian Arctic) National Park 163031 National Geographic Society Gordon and Betty Moore Foundation Grant GBMF-3781 This work is supported by Russkaya Arktika (Russian Arctic) National Park (a Federal State-Funded Organization; 57 Soviet Cosmonaut Avenue; Arkhangel’sk 163031) and the National Geographic Society (Washington, D.C.). The research was partially sponsored by the Gordon and Betty Moore Foundation Grant GBMF-3781 to FR. The funders had no role in study design, data collection and analysis, decision to publish, or preparation of the manuscript.

==============================
Cryophilic algae thrive in liquid water within snow and ice in alpine and polar regions worldwide. Blooms of these algae lower albedo (reflection of sunlight), thereby altering melting patterns (Kohshima, Seko & Yoshimura, 1993; Lutz et al., 2014; Thomas & Duval, 1995). Here metagenomic DNA analysis and satellite imaging were used to investigate red snow in Franz Josef Land in the Russian Arctic. Franz Josef Land red snow metagenomes confirmed that the communities are composed of the autotroph Chlamydomonas nivalis that is supporting a complex viral and heterotrophic bacterial community. Comparisons with white snow communities from other sites suggest that white snow and ice are initially colonized by fungal-dominated communities and then succeeded by the more complex C. nivalis-heterotroph red snow. Satellite image analysis showed that red snow covers up to 80% of the surface of snow and ice fields in Franz Josef Land and globally. Together these results show that C. nivalis supports a local food web that is on the rise as temperatures warm, with potential widespread impacts on alpine and polar environments worldwide.

Introduction

Chlamydomonas nivalis is an unicellular snow alga that has been detected worldwide within the upper snow layer in polar and alpine regions (Guiry et al., 2014) and is especially abundant in the Arctic pack ice (Gradinger & Nurnberg, 1996). In these harsh environments, C. nivalis has adapted to intense UV exposure by producing astaxanthin, a UV-screening pigment that produces a visible red hue in snow (Gorton & Vogelmann, 2003; Williams, Gorton & Vogelmann, 2003). C. nivalis spends most of its growth season in its red colored stage (Gorton & Vogelmann, 2003; Stibal et al., 2007; Williams, Gorton & Vogelmann, 2003); this coloration is visible across the snow/ice surface and can reduce albedo to 40% (c.f., fresh snow albedo of 75% (Thomas & Duval, 1995)). The lower albedo increases local temperature, promoting snow and ice melting and increasing the abundance of C. nivalis (Thomas & Duval, 1995). Through this positive feedback loop the abundance of C. nivalis amplifies snow and ice melting. C. nivalis also contributes to local carbon cycling by fixing CO2. However, if there is a red snow associated heterotrophic viral and microbial community, much of this newly fixed carbon may be released via respiration (Bardgett, Freeman & Ostle, 2008). C. nivalis-produced dissolved organic carbon (DOC) may also contribute to priming when the melt water washes into the ocean (Geller, 1986; Hamer & Marschner, 2002; Madigan, Martinko & Parker, 1997; Van Nugteren et al., 2009) and lead to increased CO2 release.

Satellite images are useful for studying remote or expansive areas that are otherwise difficult to reach and for detecting land surface changes over time. Remote sensing methods using satellite imagery are applied to a wide range of studies including urban expansion (Wang et al., 2014), agricultural land use change (Dong et al., 2015), and glacial retreats (Wei et al., 2014). Many data sets are free, easily accessible, and have adequate resolution for its purpose. Thus, remote sensing provides an ideal tool for quantifying white and red snows at large spatial and temporal extents. Here we describe the use of remote sensing to quantify white and red snows at large spatial and temporal extents. Metagenomic comparisons of white and red snows were also performed to investigate whether these snows differed in their microbial ecology.

Materials and Methods

Analysis of satellite images

Remote sensing methods were used to estimate abundances of red snow at eleven locations around the world (see Supplemental Information 1). Landsat satellite images were acquired from the USGS Earth Explorer site (http://earthexplorer.usgs.gov/) and image analysis methods were adapted from Takeuchi et al. (2006) as described in the Supplemental Information 1. Red to green reflectance band ratios with wavelengths 630–690 nanometers and 520–600 nanometers, respectively, were used to detect red snow in the satellite images. The spectral reflectance of red snow shows that it has higher reflectance in the red band than in the green band, while the spectral reflectance of white snow and ice has higher reflectance in the green band than the red band (Takeuchi et al., 2006). Therefore, red to green reflectance band ratios that are less than 1.0 are more likely to signify white snow or ice while band ratios that are greater than 1.0 are more likely to signify red snow or ice.

ArcGIS version 10.2 was used to calculate the reflectance band ratios. Previous research indicates areas with reflectance band ratios >1.02 are bright red when observed in the field (Takeuchi et al., 2006). For this analysis, areas with reflectance band ratios greater than 1.0 were considered to have a significant amount of red snow because such values have been shown to have an algal cell volume of 100 ml m−2 (Takeuchi et al., 2006). Using the positive linear correlation between algal cell volume biomass and reflectance band ratio, it was assumed that the higher the reflectance band ratio, the higher the algal cell volume biomass. With this in mind, the reflectance band ratios were divided into five categories for optimal visualization of various levels of concentrations of red snow (Table S1 and Fig. 1B).

Figure 1 Changes in red snow through time.

(A) A time series comparison of the percentage of total snow or ice that is covered with algae at selected alpine and polar regions throughout the world, according to data derived from satellite images. (B) A time series comparison of the total area of snow and sea ice, total algal biomass, and percentage of total snow that is covered with algae within the map extent near Nansen Island, Franz Josef Land, for years 1986, 2002, 2006 and 2015. The colored time series shows spatial distribution maps of algal densities of the Nansen Island area in Franz Josef Land.

Algal biomass

To estimate the algal biomass for each location, the surface area belonging to each reflectance band ratio category was multiplied by the mean algal biomass of that category. Although the extent of the area of interest is the same for all three images, they have varying amounts of surface area where red snow can exist due to shifts in snow and ice coverage. Therefore, in addition to the total algal biomass, the total area of snow coverage and the percentage of the total area of snow that was covered with different abundances of red algae were calculated. A pixel was categorized as snow if its normalized difference snow index (NDSI) was greater than 0.4 and, to mask out water, if its near-infrared reflectance value was greater than 0.11 (Sibandze et al., 2014). The number of pixels that meet these conditions was multiplied by the area of the pixel to get the total area of ground covered by snow/ice. To calculate the percentage of the total area of snow that is covered with algae, the total area with each algal abundance level was divided by the total area of snow coverage.

Metagenomic sequencing

Permitting for this work was from the Russian Federation (Ministry of Education and Research #71; June 3, 2013). Red snows were sampled on Nansen Island (“Nansen”) and Greely Island (“Greely_1” and “Greely_2”) of Franz Josef Land. Red snow samples were examined with microscopy to confirm the presence of C. nivalis based on morphology (Muller et al., 1998). Three red snow samples of ∼15 L were collected, melted, and passed through a 0.22 µm sterivex filter. Greely_1 and Greely_2 represent two different sterivex filters that were both extracted from the same homogenized sample. Total DNA was extracted in the field using the Soil DNA Isolation kit with a custom vacuum manifold (cat# 26560; Norgen BioTek Corp.,Thorold, Ontario, Canada). From the total DNA, a NexteraXT library kit was used to prepare DNA libraries for sequencing on the Illumina MiSeq. The Nansen, Greely_1, and Greely_2 libraries had 135,749 reads, 86,932 reads and 47,507 reads, respectively (see Table S2 for MG-RAST IDs to obtain unfiltered data). Each metagenome was passed through the following quality control pipeline. PrinSeq was used to quality filter reads below 100 bp in length and below an average quality score of 25, and to remove duplicates and sequence tags (Schmieder & Edwards, 2011b). Reads assigned as human were removed using DeconSeq (Schmieder & Edwards, 2011a). Post quality control, the Nansen library contained 121,455 reads, Greely_1 contained 69,918 reads, and Greely_2 contained 40,344 reads. Seven publicly accessible white snow metagenomes from Svalbard glaciers (a.k.a., ‘white snow’ throughout manuscript) sampled April through June were downloaded from MG-RAST (see Table S2 for MG-RAST IDs), and reads were quality filtered using the same pipeline as the Franz Josef Land red snow libraries (Maccario, Vogel & Larose, 2014). Metagenomes were analyzed using KEGG and M5NR databases within MG-RAST version 3.3 (Meyer et al., 2008). The red snow and white snow libraries were compared to the KEGG database to assign reads to KEGG pathways (e-value <1 × 10−5; >60% identity; >15 aa minimum alignment length). Estimations of taxonomic composition of communities were based on translated comparisons to the non-redundant protein database M5NR (e-value <1 × 10−5; >60% identity; >15 aa minimum alignment length). The dataset was normalized to ensure similar numbers of reads were used for each sample, and then raw read counts were log transformed. Statistical differences between red snow and white snow in the numbers of reads assigned to KEGG pathway groups were identified by ANOVA. Multivariate statistics were performed in R using the vegan (Dixon, 2003), clustsig and the stats packages. The adonis function was used to compare metagenome compositions; vegdist was used to generate distance matrices; simprof was used to cluster metagenomes based on similarity; and prcomp was used to perform Principal Component Analysis.

Results and Discussion

Detection of red snow in a global sample of satellite images

Satellite images with spectral reflectance data were used to approximate snow and ice cover, and red algae abundance (Takeuchi, 2009; Takeuchi et al., 2006) over several years in Franz Josef Land, as well as eleven other regions of United States, Canada, Greenland, Norway, Austria, India, and New Zealand (Fig. S1). Red snow was detected at all eleven locations in all the years (Fig. 1A). The total area of snow and ice were lowest in the most recent year (2013, 2014 or 2015, depending on the location; Fig. S2; Greenland was the exception to this trend). At least 50% of the total snow/ice area was covered with red algae for the most recent year analyzed (Fig. S2; exception New Zealand and Franz Josef Land). In seven of the locations, over 80% of the total snow and ice fields were covered in red algae in the most recent year analyzed (Fig. S2).

A walking transect from sea level to the glacier on Nansen Island in Franz Josef Land was performed in August 2013 (to be described in a separate manuscript). Therefore, this region was targeted for more detailed analysis. Around and on Nansen, the total red snow algal biomass increased by 124% from 1986 to 2002 and by 15% from 2002 to 2006, then decreased by 63% from 2006 to 2015 (Fig. 1B). These changes in algal cover co-occurred with a total decline in the snow and ice cover (Fig. 1B). Visual inspection of the snow and ice on Nansen Island in August of 2013 confirmed the presence of red colored snow and microscopy of red snow samples showed C. nivalis cells. Taken together, these results show that even as total snow and ice cover declines, red snow cover is still highly prevalent or increasing both in Franz Josef Land and other alpine/polar regions.

Microbes present in white snow and red snow

For metagenomic sequencing, red snow samples were taken from Nansen and Greeley Islands, respectively. Seven white snow metagenomes from Svalbard glaciers were also downloaded and analyzed for comparison (see ‘Methods’ & Table S2 for MG-RAST ID numbers). The genus-level taxonomic compositions of white snow and red snow were significantly different (ADONIS; F = 4.567; p = 0.007). When samples were clustered according to their taxonomical similarities, one red snow sample taken at Greely Island grouped with a Svalbard glacier sample; otherwise the red snow and white snow samples clustered separately (Fig. S3). This indicates minimal overlap in microbial composition at the genus level.

Community DNA sequences were further compared using multivariate analyses with the top 10 most variable taxa (Fig. S4). The first two principal components explained 70% of the between-sample variation in microbial community members. The first principal component described red snow as having higher abundances of species from the bacterial genera Pseudoalteromonas, Alteromonas, Vibrio, and Pedobacter, whereas white snow had higher abundances of species from the eukaryotic genera Aspergillus and Neurospora, as well as the bacterial genera Nostoc, Bacillus and Spirosoma. Red snow had greater overall abundances of Bacteria and viruses (Fig. 2A) and a lower abundance of Eukaryotes (Fig. 2A). The bacterial communities associated with red snow have also been observed in an alpine region (Thomas & Duval, 1995) and are probably supported by photosynthate from the C. nivalis. Evidence also suggests that bacterial cells may physically attach to the outer mucilaginous coating of C. nivalis in red snow, forming an Arctic holobiont (Bordenstein & Theis, 2015; Remias, Lutz-Meindl & Lutz, 2005; Thomas & Duval, 1995).

Figure 2 Abundances of microbes in red snow and white snow samples.

(A) Abundances of viruses, Bacteria and Eukaryotes in samples from red snow and snow communities. The y-axis shows abundances after normalizing and standardizing raw read counts to ensure cross-sample comparisons are valid. (B) Bar plots showing abundances of two Eukaryotic phyla found in red snow and snow communities. Chlorophyta is the phylum that contains the genus Chlamydomonas.

The metagenomes were also used to verify the presence of Chlamydomonas in snow samples (Fig. S5). Of the sequence reads assigned to Eukaryotes, the proportion of reads assigned to the Chlamydomonas-containing phylum Chlorophyta was higher in red snow than white snow (Fig. 2B). Conversely, the proportion of reads assigned to the fungal phylum Ascomycota was higher in white snow (Fig. 2B).

Functional capabilities of microbial communities in red snow and white snow

The metagenomes were also analyzed for potential metabolic functions. The functions encoded by the red and white snow samples clustered into 8 significant groups, with the red snow samples from Greely and Nansen Islands forming a significant cluster (Fig. S6). Four white snow samples formed a cluster and the remaining white snow samples clustered individually. Multivariate analysis of the top 10 most variable functions showed that the first two principal components explained 82% of the variation in the abundances of functional pathways (Fig. S7). The first component (70% of the variation) showed that the red snow had higher abundances of genes involved in membrane transport, carbohydrate metabolism, nucleotide and amino acid synthesis/degradation, and energy metabolism. White snow communities were shifted toward cell growth and death, folding sorting and degradation, transcription, transport and catabolism pathways and pathways annotated as important in neurodegenerative diseases (i.e., mitochondrial functions in Eukaryotes).

In order to examine whether microbial communities in red snow encoded completely different functional capabilities from those in snow, the numbers of reads assigned to all KEGG pathways were compared using a matrix of Bray-Curtis dissimilarities. Overall the abundances of level 1 KEGG pathways were not significantly different between red snow and white snow (ADONIS; F = 2.135; p = 0.12). However, separate analyses that compared individual pathways (at level 2) between red snow and white snow identified several pathways as significantly different, including pathways related to sugar biosynthesis and metabolism and energy metabolism. Red snow communities had higher abundances of genes that encode lipopolysaccharide biosynthesis and peptidoglycan biosynthesis (Fig. 3A). Red snow also had a higher proportion of reads assigned to oxidative phosphorylation, methane metabolism, carbon fixation in photosynthetic organisms and carbon fixation pathways (Fig. 3B). White snow had higher relative abundances of genes that encode glycan biosynthesis and related pathways such as GPI-anchor biosynthesis, other types of O-glycan biosynthesis and various types of N-glycan biosynthesis (Fig. 3A).

Figure 3 Functional pathways in red snow and white snow.

(A) Bar plots showing functional pathways that were statistically significantly different in abundance between red snow and snow. (B) Bar plots depicting energy metabolism pathways and their abundances in red snow and snow.

Taken together, these results support the hypothesis that red snow communities include primary producers with a large, heterotrophic community including viruses. These red snow communities are photosynthesizing and fixing carbon, and also metabolizing methane, processes that could accelerate snow melting. In contrast, white snow communities appear to be dominated by fungi, maybe eating refractory organic carbon delivered with the snow (Clarke & Noone, 1985; Rosen, Novakov & Bodhaine, 1981; Thevenon et al., 2009). These white snow communities are lacking the signatures of primary productivity.

Figure 4 Model of red snow microbiology.

Model of C. nivalis in white and red snows. (A) Several microbial communities that are found in white snow. Sunlight promotes astaxanthin expression in C. nivalis, turning the snow to red and promoting community metabolism shifts through stimulation of heterotrophic metabolism. The C. nivalis blooms, albedo is decreased and local snow and ice melts at a faster rate (B).

Conclusions

Microbiology of snow and ice fields has a long history, including a reference to red snow by Aristotle. However, until now we have not had the tools to determine the full extent and makeup of these communities. Here we use a combination of satellite and metagenomic approaches to show that red snow covers up to 80% of the examined ice and snow fields. Metagenomics of red snow from Franz Josef Land, one of the most remote polar land masses in the world, show that these communities support a full food web ranging from algae to heterotrophic microbes to viruses. Because of the reduced albedo associated with these communities, red snow creates a positive feedback loop that increases its abundance while simultaneously melting ice and snow (Fig. 4). In addition to the direct effects on sunlight absorbance, the heterotrophic activity (including viral lysis) will increase local temperatures. Together, these effects may significantly increase ice and snow melting in the Barents Sea region that is already one of the fastest-warming regions on earth. Projections for global red snow coverage and its influence on warming patterns should be investigated further.

Supplemental Information

Supplemental Information 1 Supplementary methods

Click here for additional data file.

Figure S1 Satellite image sites

Black points indicate the locations analyzed for red snow content. Harding Ice Field, United States (1); Olympic National Park, United States (2); Rocky Mountains, Canada (3); Sierra Nevada Mountain Range, United States (4); Glacier National Park, United States (5); Mittavikkat Glacier, Greenland (6); Svalbard Archipelago, Norway (7); Grossglockner Mountain, Austria (8); Nansen Island, Franz Josef Land (9); Himalaya Mountain Range, India (10); Mt. Cook, New Zealand (11).

Click here for additional data file.

Figure S2 Time series comparisons

A time series comparison of the total area of snow and sea ice, total algal biomass, and percentage of total snow that is covered with algae at ten different study sites.

Click here for additional data file.

Figure S3 Taxonomic similarities-Clustering

Clustering of snow and red snow communities based on similarities in taxonomic composition. Colored branches indicate significant clusters, with each color representing one cluster. Labels with ‘SVN’ correspond to white snow samples and ‘Greely 1’, ‘Greely 2’, and ‘Nansen’ correspond to red snow samples.

Click here for additional data file.

Figure S4 Taxonomic similarities-PCA

Plot from principal components analysis of red snow and snow community taxonomic composition. Arrows indicate over-representation of certain taxa (red text) in particular red snow or snow communities (black text). Labels are described in the legend for Fig. S3.

Click here for additional data file.

Figure S5 Chlorophyta

Bar plots showing genus level comparisons of reads assigned to Chlorophyta. The y-axis shows the proportion of reads assigned to the phylum Chlorophyta for each genus. Red bars indicate data from red snow samples and khaki bars indicate data from snow. Snow metagenomes did not contain any reads from the following genera: Scenedesmus, Pyramimonas, Dunaliella, Prototheca, and Pseudendoclonium.

Click here for additional data file.

Figure S6 Functional similarities-Clustering

Clustering of snow and red snow samples based on similarities in functions encoded by each community sample. Colored branches indicate significant clusters, with each color representing one cluster. Labels are described in the legend for Fig. S3.

Click here for additional data file.

Figure S7 Functional similarities-PCA

Principal components analysis of red snow and snow community functions. Arrows indicate over-representation of certain functional categories (red text) in particular red snow or snow communities (black text). Labels are described in the legend for Fig. S3.

Click here for additional data file.

Table S1 Predicting algal abundance

The relationship between reflectance band ratio values, C. nivalis biomass, and the proposed level of algae abundance, extrapolated from a positive linear correlation between reflectance band ratio and algal biomass that was shown in previous research (Takeuchi et al., 2006).

Click here for additional data file.

Table S2 Metagenomic samples

Click here for additional data file.

Supplemental Information 2 Landsat IDs

This file lists the Landsat IDs used for satellite image analysis of snow/ice/red snow abundances. Images can be acquired freely at: http://earthexplorer.usgs.gov/.

Click here for additional data file.

The authors are grateful to Alexander Chichaev, Roman Seliverstov, Pavel Terekhov, Andrew Terekhov, and Sergey Kononov for keeping us alive above water and Dave McAloney for underwater operations. The authors also thank Yuri Gavrilov (INTAARI, St. Petersburg, Russia) and Paul Rose (Royal Geographic Society) for logistical help, as well as the captain and crew of the M/V Polaris.

Additional Information and Declarations

Competing Interests

Author Contributions

Field Study Permissions

DNA Deposition

Data Availability

The authors declare there are no competing interests.

Nao Hisakawa and Eric R. Hester performed the experiments, analyzed the data, wrote the paper, prepared figures and/or tables, reviewed drafts of the paper.

Steven D. Quistad conceived and designed the experiments, performed the experiments, wrote the paper, prepared figures and/or tables, reviewed drafts of the paper.

Daria Martynova performed the experiments, contributed reagents/materials/analysis tools, wrote the paper, prepared figures and/or tables, reviewed drafts of the paper.

Heather Maughan analyzed the data, wrote the paper, prepared figures and/or tables, reviewed drafts of the paper.

Enric Sala performed the experiments, contributed reagents/materials/analysis tools, reviewed drafts of the paper.

Maria V. Gavrilo performed the experiments, contributed reagents/materials/analysis tools.

Forest Rohwer conceived and designed the experiments, performed the experiments, contributed reagents/materials/analysis tools, wrote the paper, reviewed drafts of the paper.

The following information was supplied relating to field study approvals (i.e., approving body and any reference numbers):

Permitting for this work was from the Russian Federation (Ministry of Education and Research #71; June 3, 2013).

The following information was supplied regarding the deposition of DNA sequences:

MG-RAST id numbers: 4614610.3, 4614611.3, and 4614609.3.

The following information was supplied regarding data availability:

MG-RAST: http://metagenomics.anl.gov/linkin.cgi?project=12309.

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
