# Peer review of "Metagenomic and satellite analyses of red snow in the Russian Arctic"

_PeerJ, doi:10.7717/peerj.1491_

## Round 0.1 · original submission · Minor Revisions

Please address all the reviewers' comments, including making a note in the text that MG-RAST id numbers are provided as accession numbers in supplementary table 2 for the datasets described.

Additionally, please address the following:

1. Line 11, should be 'lower' not 'lowers'
2. Line 111, please clarify if 50% was covered with algae or red algae
3. Line 143, should be 'supported' not 'support
4. Line 188, insert 'the' as so '..land masses in the world..'
5. Please take a careful look at the labels of supplementary figure 5. There should be a space between 'red' and 'snow' in the color key. The y-axis should probably say 'Proportion of reads assigned to Chlorophyta.' The figure legend line 2 should say 'The y-axis..' rather than 'They y-axis..'

Reviewer 1 ·

Basic reporting

This manuscript by Hisakawa et al analysed the microbial communities mainly algae inhabiting the arctic snow using metagenomic and satellite approaches. Authors aim that albedo is reduced due to inhabitation of microbes on snow/ice. The MS provides a good comparison with other regions. Some points which needs to be edited are:
1. In line, 90 FJL is not described (should be abbreviated in the beginning).
2. Throughout the text, the white snow is represented sometimes as 'snow' and sometimes as 'white snow'. It should be consistent.
3. Introduction mainly talks about C.nivalis, more background information about the approaches used in the study and their importance would be good.

Experimental design

1. Why sequencing libraries were named the Nansen and Greeley? Are these two different samples from two different sites? Details of site information required.
2. Biological replicates would be appreciated, Are Nansen and Greeley replicates?

Validity of the findings

The Conclusions section is rather shallow. Certain speculations on the red snow communities include primary producers in contrast to white snow communities dominated by fungi need to be discussed in detail.

Reviewer 2 ·

Basic reporting

According to the PeerJ Policies, the manuscript should include accession numbers for all the Metagenomic dataset. Accession numbers should be provided for the Greely_1, Greely_2 and Nansen dataset but also the white snows sampled used in the investigation.
In the deposit of your data, please indicate the status of the data (filtered or not).

As you have used Kegg and M5NR to demonstrate lack or difference between the metagenomic samples - and as Kegg and M5NR are evolving database, it appears that in the future new annotations can change slightly some of your computations - so please, give indications about the database you used (release date, size, number of entries - what do you think is valuable to describe the database you are using)

Experimental design

You have used Bray-Curtis measure to expose some differences between abundance on Kegg pathway. May I suggest to add an example on how you are using this measure and the size of the set where you are performing the Bray-Curtis measure?

Validity of the findings

No Comments

Additional comments

Nice job on a very interesting subject.

---

## Round 0.2 · accepted · Accept

Thank you for addressing each of the comments that were raised by myself and the reviewers.